# Unlocking the Therapeutic Potential of Medicinal Plants for Alzheimer's Disease: Preclinical to Clinical Trial Insights

**Kushagra Nagori** [1], **Kartik T. Nakhate** [2,*], **Krishna Yadav** [3], **Ajazuddin** [1] and **Madhulika Pradhan** [4,*]

[1] Rungta College of Pharmaceutical Sciences and Research, Kohka-Kurud Road, Bhilai 490024, India; kushagranagori13@gmail.com (K.N.); write2ajaz@gmail.com (A.)
[2] Department of Pharmacology, Shri Vile Parle Kelavani Mandal's Institute of Pharmacy, Dhule 424001, India
[3] Raipur Institute of Pharmaceutical Education and Research, Raipur 492010, India; drkrishnayadav2022@gmail.com
[4] Gracious College of Pharmacy, Abhanpur, Raipur 493661, India
[*] Correspondence: kartiknakhate@gmail.com (K.T.N.); madhulika.pradhan1@gmail.com (M.P.)

**Abstract:** Alzheimer's disease (AD) is a progressive, multifactorial, and unremitting neurodegenerative disease characterized by memory loss, personality changes, and cognitive impairment. It has become more prevalent in recent years. Therefore, understanding the pathophysiology of AD and developing efficient therapeutic strategies are essential. Moreover, the progression of the disease is unaffected by the pharmaceutical approaches discovered to date. Additionally, the failure of over 200 potential drug candidates in clinical trials over the past decade suggests the complexity and difficulty of both the disease and its underlying causes. Therefore, research focused on medicinal plant-based natural products in the search for novel neuroprotective therapeutic candidates for AD is essential. Indeed, several scientific investigations have demonstrated the efficacy of many medicinal plants and their principal phytochemicals in the treatment of AD. This review article covered the pathophysiological mechanisms of AD, the necessity for natural products as anti-AD treatments, and the most recent preclinical studies revealing the function of neuroprotective medicinal plants and their bioactive compounds in the effective management of AD. In addition, the review also presents clinical trial data of promising anti-AD formulations/agents of plant origin. Revealing recent findings and highlighting the clinical trial data related to the development of new treatments for AD would promote further research in this field and pave the way for the development of more effective and safe treatments for this debilitating disease.

**Keywords:** neuroprotective; Alzheimer's disease (AD); medicinal plants; clinical trial

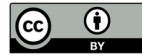

## 1. Introduction

The most prevalent form of dementia worldwide is Alzheimer's disease (AD), a progressive neurological condition with a high social and caregiver cost. It is an irreversible neurodegenerative disorder marked by occasional unusual behavior, memory loss, and cognitive impairment. An increase of more than 145% in deaths attributed to AD was observed between 2000 and 2019, and it would likely be exacerbated due to the pandemic of COVID-19 [1,2]. As the world's population ages, the number of people with dementia is currently projected to become over 40 million, and this number is expected to keep growing, doubling every 20 years [3,4]. Moreover, around 58 million individuals worldwide are estimated to be suffering from AD as of 2021, which is anticipated to increase to 88 million by 2050. The survey data also reveal that approximately 6.2 million people in the United States are suffering from AD [5,6].

AD is a neuronal cell disorder that generates progressive dysfunction of neurons due to the abnormal deposition of amyloid β (Aβ) and hyperphosphorylated tau proteins, genetic susceptibility, and increased oxidative stress. Eventually, this may lead to cognitive

impairment and impact the function of the central and peripheral nervous systems [7,8]. Normally, synthesized Aβ released into the extraneuronal space, where microglia and astrocytes clear it. In AD, increased Aβ production or impaired Aβ degradation/clearance results in its accumulation. Tau is primarily expressed in neurons and is significantly regulated by a variety of post-translational modifications (PTMs). Intraneuronal tau accumulation can be caused by abnormal PTMs, liquid–liquid phase separation (LLPS), and pathogenic tau seeds via many pathways. Tau pathology can spread during disease progression, and glial cells play an essential role in seeding and dispersal. Aβ and other forms of Aβ aggregates, in conjunction with tau accumulation, can result in neuronal dysfunction, glial activation, and eventually neuroinflammation; these events are regulated by different receptors expressed in neurons, microglia, and astrocytes [9].

The use of acetylcholinesterase inhibitors (AChEI), a class of drugs created especially for AD, began in the late 1990s [10,11]. Their usage is supported by the neurochemical discovery that Alzheimer's individuals' brains exhibit a decrease in the neurotransmitter acetylcholine. The benefit of these medications is fairly limited, and the studies have reported that donepezil, rivastigmine, and galantamine have the potential to marginally improve cognition, ability to perform routine activities, and behavior in cases with mild-to-moderate AD for intervals of 6 to 18 months [12]. Memantine, which blocks glutamate at the NMDA receptor, is the other disease-specific medication. Notably, none of the medications currently approved for use in AD have been shown to have a disease-modifying effect; as a result, they have no impact on the course or severity of the condition [13–15]. Moreover, unfavorable side effects related to their administration compromise the effectiveness of the currently available conventional therapy against diseases. Another element limiting their utilization is their exorbitant cost [16]. In the context of these problems related to conventional treatment, the ancient practice of using traditional herbal remedies has grown in popularity as the most preferred alternative to conventional drugs to address these drawbacks of synthetic drugs [17,18].

People are inclined toward herbal medicine for many reasons; as science and technology advance, herbal medicines' quality, safety, and effectiveness improve. Moreover, the belief is that chemically produced goods are inferior to herbal products [19]. Additionally, consumers believed that herbal medicines are cost-effective and more effectively treat some ailments than some conventional medications [20,21]. New treatment approaches based on medicinal plants have recently come into focus because AD has become a concern to public health, and the commonly used synthetic drugs have unfavorable side effects. A wide range of medicinal plants and their bioactive plant compounds and crude extracts seem to have significant potential to treat and prevent AD [22,23]. Additionally, to develop prospective medication candidates to treat neurologic dysfunctions like AD, many drug discovery initiatives have been recently applied to medicinal plants. In fact, according to estimates from the World Health Organization, about 3.4 billion people in developing nations prefer plant-based medications [24,25].

This review summarizes the pathophysiology of AD and highlights the role of medicinal plants in the treatment of AD. Additionally, this review specifically draws attention to the recent developments in preclinical in vitro and in vivo trials to demonstrate the efficacy of herbal medicine in increasing effectiveness or combating AD.

## 2. Methodology

The purpose of this research is to outline the current understanding of the pathogenesis of AD and to assess the neuroprotective potential of several medicinal plants against AD. Using electronic databases like PubMed, Scopus, Web of Science, and Google Scholar, this section explains how we found relevant articles, extracted relevant data, and analyzed the data. We conducted our search using the following key phrases: Alzheimer's disease, neuroprotection, medicinal plant, and the full names of the plants we have included in the study. Only papers published in English between 1990 and 2023 were considered for this search. Data were retrieved and summarized into a table after the search and screening

were performed. Name of medicinal plant, plant parts used, active components, preclinical research, clinical studies, and purported neuroprotective effects were all listed in a table. Finally, the results were categorized based on the neuroprotective effects observed in the study.

### 3. Pathophysiology of Alzheimer's Disease

AD is the most common form of dementia and one of the biggest healthcare concerns of the 20th century. It is an irreversible neurodegenerative disease characterized by memory and cognitive impairment [26,27]. Additionally, people in the age group of 65 and above are more prone to AD. Senile plaques (SPs) and neurofibrillary tangles (NFTs) are the key pathological markers of AD. Moreover, significant evidence demonstrates that amyloid peptide (Aβ) play a crucial role in the pathophysiology of AD [26,28]. Therefore, the main pathological events in AD are aberrant cleavage of amyloid precursor protein (APP), which leads to the buildup of Aβ plaques in the extracellular space, and hyperphosphorylation of tau protein, which leads to the development of intracellular neurofibrillary tangles (NFTs). Interestingly, the studies indicate that the increase in Aβ triggers tau pathology followed by neuronal death and, eventually, the disease [29]. The overview of the pathophysiology of AD is depicted in Figure 1.

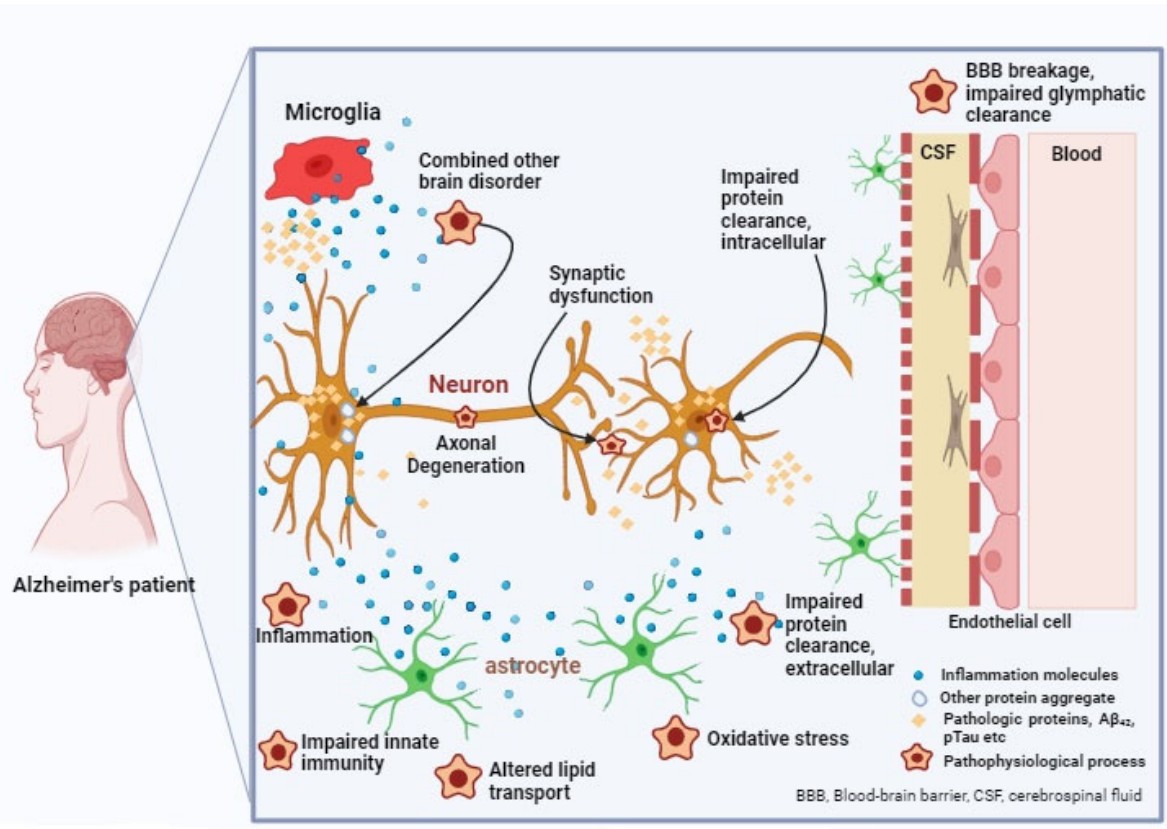

**Figure 1.** Outline of pathophysiological process in Alzheimer's disease.

However, the primary building block of amyloids or SPs is a peptide fibril made of Aβ, which is highly insoluble and proteolysis resistant. The pathology and treatment of AD are typically based on several putative pathological mechanisms [30]. For instance, the tau hypothesis, amyloid cascade hypothesis, excitotoxicity, cholinergic hypothesis, autophagy, neuroinflammation, and oxidative stress are a few of the suggested mechanisms of AD, which are discussed in the following sections.

### 3.1. Amyloid Cascade Hypothesis

The Aβ peptide accumulation and deposition in the brain parenchyma was hypothesized to play a significant role in understanding AD etiology in 1992, giving rise to the amyloid cascade hypothesis [31]. AD is characterized neuropathologically by neuritic plaques in the brain. They are produced as a result of extracellular Aβ deposition and aggregation. Aβ is derived from the sequential cleavage of APP by β-secretase and γ-secretase. β-site APP-cleaving enzyme 1 (BACE1) is the major, if not the only, β-secretase in vivo and is required for Aβ synthesis. The regulation of APP processing is a prominent focus of AD pathogenesis research. The APP trafficking mechanisms and cleavage enzymes are complex. Transporting APP and secretases into the same subcellular organelles promotes interaction and APP processing [32,33]. The amyloid cascade hypothesis is further supported by the mutation of the APP, presenilin1, and two genes (PSEN1 and PSEN2), which is responsible for the change in proteolytic cleavage of APP, increasing the ratio of longer self-aggregating Aβ peptides. This mutation was found in families with early-onset AD, and a significant portion of Down's syndrome patients showed an early indication of AD. Although the amyloid cascade hypothesis has a wealth of evidence, it has repeatedly been demonstrated that Aβ accumulation and consecutive deposition are unrelated to neuronal loss or cognitive decline. The fact that many patients have amyloid plaque burden but no cognitive impairment symptoms suggested that the hypothesis may not be totally accurate [15,34,35].

### 3.2. Tau Hypothesis

Tau is a microtubule-associated protein that promotes microtubule polymerization and stabilization in the cell cytoskeleton. Therefore, tau phosphorylation and aggregation are the key causes of neurodegeneration in AD [36]. According to the tau hypothesis, tau tangle pathology develops prior to the formation of Aβ plaque because hyperphosphorylation of tau leads to clumping into paired helical filaments (PHFs) and NFTs and detach it from microtubules. Therefore, tau phosphorylation and aggregate formation are the key factors responsible for neurodegeneration in AD [37–39].

### 3.3. Cholinergic Hypothesis

The cholinergic hypothesis was the first theory put forth to explain AD. The function of acetylcholine in learning and memory was recognized, and cholinergic abnormalities in the brains of AD patients led to the emergence of the cholinergic hypothesis [40,41]. The Food and Drug Administration has currently approved three AChEIs (rivastigmine, donepezil, and galantamine) for treating mild-to-moderate AD. The investigations demonstrate that AChEIs can temporarily alleviate some symptoms but cannot permanently treat this condition. Moreover, the development or repurposing of drugs that can target many aspects of the disease, such as risk factors, mechanism-based versus non-mechanism-based strategies, symptomatic treatments, and lifestyle modifications, should be the focus of novel AD strategies [42,43]. Interestingly, most research into AD drugs has focused on finding compounds that specifically choose one or more of the postulated pathophysiologic mechanisms of AD [15].

### 3.4. Autophagy and Neuroinflammation

Autophagy serves as a natural cellular stress response, eliminating detrimental or unnecessary substances from cells by utilizing the lysosomal function [44]. Autophagy allows for the recycling of nutrients to give energy during hunger, but it is also engaged in other aspects of cellular stress management in complex multicellular organisms. Indeed, autophagy functions as a quality control mechanism, maintaining cellular homeostasis by destroying aggregate-prone proteins and damaged organelles [45]. Furthermore, autophagy adversely affects several essential immune response components, playing an important role in counteracting inflammation [46]. Studies covering the entire genome have

revealed that proteins engaged in endocytic trafficking, such as PICALM/CALM (phosphatidylinositol-binding clathrin assembly protein), play a role in AD pathology by influencing autophagy and affecting the removal of tau protein, both in vitro and in vivo [47]. In line with these findings, the downregulation of beclin 1 (BECN1) increases the aggregation of Aβ plaques and tau aggregates in AD [48]. Furthermore, the pathological forms of Aβ and αS disrupt the autophagic process, leading to a detrimental cycle of adverse events [49]. Inflammation is defined as a complex series of events caused by irritation, injury, or infection of a specific tissue, which may eventually develop into a cleansing response by innate immune system components and even trigger an adaptive immune response specific to the damaging agent. Neuroinflammation is a prevalent hallmark of various neurodegenerative illnesses and is usually present whenever an injury is put on the nerve tissue. The function of neuroinflammation in aging and AD has been extensively studied during the last two decades. Much has been revealed about the cellular and molecular participants in the brain parenchyma, but also at the level of the brain's barriers and at the periphery, specifically in circulation and organs with a strong immune-related activity [50]. Microglia, the central nervous system's resident immune cells, play an important role in regulating brain inflammation and have recently attracted attention as critical regulators of neurodegenerative processes. Autophagy is also known to be an important mediator of inflammation. The degradation of the NOD-, LRR-, and pyrin domain-containing protein 3 (NLRP3) inflammasome, an intracellular complex responsible for the generation of inflammatory cytokines interleukin1b and interleukin-18, is an example of autophagic regulation of inflammation. The data imply that autophagy regulates the NLRP3 inflammasome, and a deficiency in autophagy directly leads to inflammation. Furthermore, inflammatory cytokines can have an effect on autophagy. In a mouse model of AD, treatment of interferon gamma enhances microglial clearance of Aβ [29].

### 3.5. Oxidative Stress

Oxidative stress is characterized by a disparity between the generation and elimination of oxidants, resulting from inadequate functioning of antioxidant defenses. Consequently, there is an increased accumulation of reactive species originating from oxygen and nitrogen. The primary outcome of this imbalance is the oxidative alteration of lipids (lipid peroxidation), proteins, and nucleic acids [51]. In the context of Aβ production, multiple investigations have revealed that oxidative stress leads to the formation of SPs by lowering α-secretase activity while increasing both β- and γ-secretase activities [52]. Additionally, the buildup of oligomers exacerbates oxidative stress. It has been observed that Aβ can induce a concentration-dependent accumulation of reactive oxygen species (ROS) and boost their generation by directly activating NADPH oxidase [53].

### 3.6. Tau Truncation

During the progression of AD and other dementias associated with tauopathy, tau truncation takes place in the early stages. Truncation, among tau post-translational modifications, is particularly involved in the onset/progression of AD due to its capacity to induce both misfolding/aggregation and neurodegeneration [54]. Tau cleavage increases glycation and ubiquitination, two major pathological protein changes that occur as PHF mature into late-stage filamentous inclusions over time. Truncation, on the other hand, results in tau fragments, which can cause neurodegeneration independent of the pro-aggregative pathway(s) and along a fragment-specific pattern [55]. The common mechanism of neurodegeneration is depicted in Figure 2.

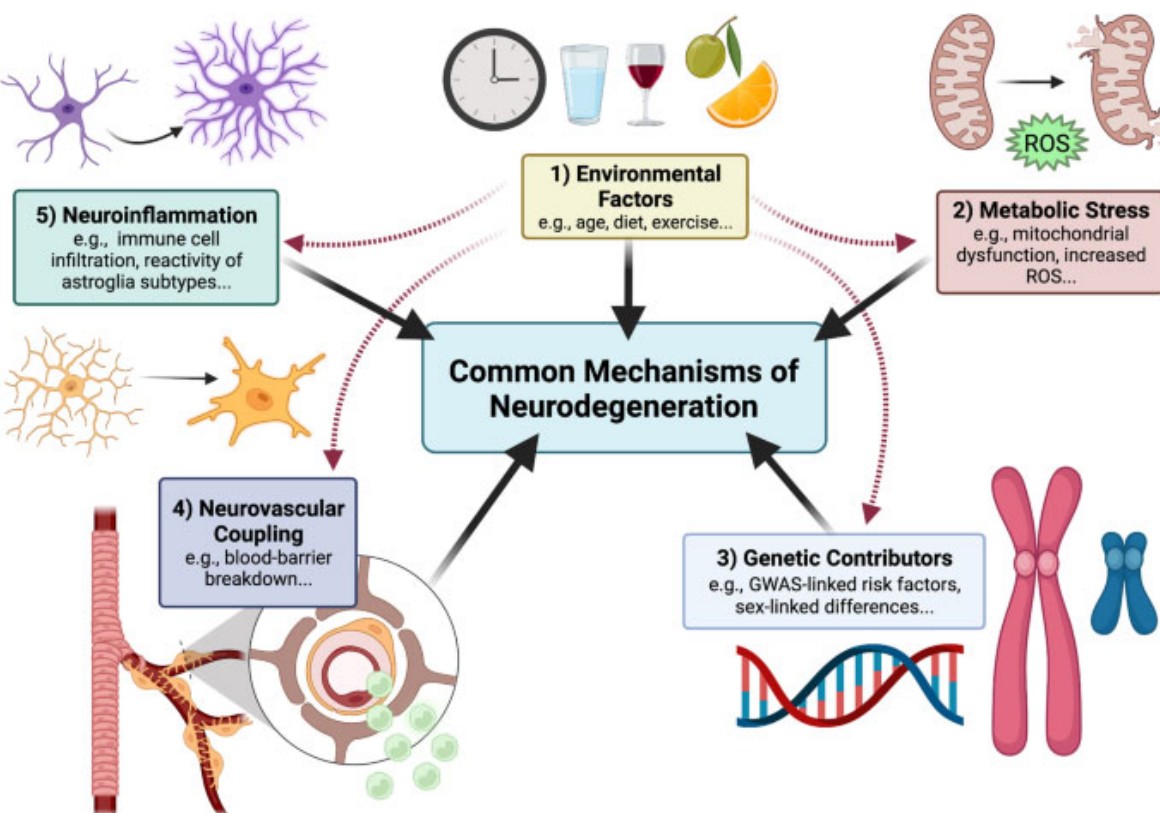

**Figure 2.** Common mechanisms of neurodegeneration (Adopted from [56]).

## 4. Neuroprotective Potential of Medicinal Plants against Alzheimer's Disease

Medicinal plants have been used to treat diseases worldwide since the dawn of mankind. Natural products have drawn much interest and contributed to the development of new drugs. Medicinal plants can play a role in supporting the health of the neuronal neurotransmitter systems in the brain. Neurotransmitters are chemical messengers that transmit signals between nerve cells, and an imbalance in these neurotransmitters can contribute to various neurological and mental health disorders. Several medicinal plants have been studied for their potential to influence neurotransmitter levels and function [57]. Plant extracts may be superior over single drugs since they can affect multiple targets at once, which makes them a more potent and novel treatment alternative for AD. Alkaloids, flavonoids, and phenolic acids are among the essential secondary metabolites of plants that play a significant role in preventing neurodegeneration in AD [17,58,59]. The antioxidant-rich plants may assist in reducing the pathogenesis of neurological symptoms by preventing oxidative stress, which is known to be one factor that accelerates AD progression. Indeed, due to the significant effect of medicinal plants, without noticeable side effects and owing to their vast array of chemical diversity, scientific interest is being developed by researchers to use them for the treatment and prevention of AD [60,61]. However, only a few herbal remedies have been studied in clinical testing despite many of them showing promising results experimentally. The medicinal plants that have shown effectiveness in treating AD include *Panax ginseng* (PG), *Curcuma longa* (CL), *Centella asiatica* (CA), *Ginkgo biloba* (GB), *Glycyrrhiza glabra*, *Withania somnifera* (WS), *Bacopa monnieri* (BM), *Tinospora cordifolia* (TC), and *Convolvulus pluricaulis* (CP). These plants are exclusively discussed in this review. A few plants with anti-AD and other pharmacological activities are shown in Figure 3. This review provides new insights into the treatment of AD [15].

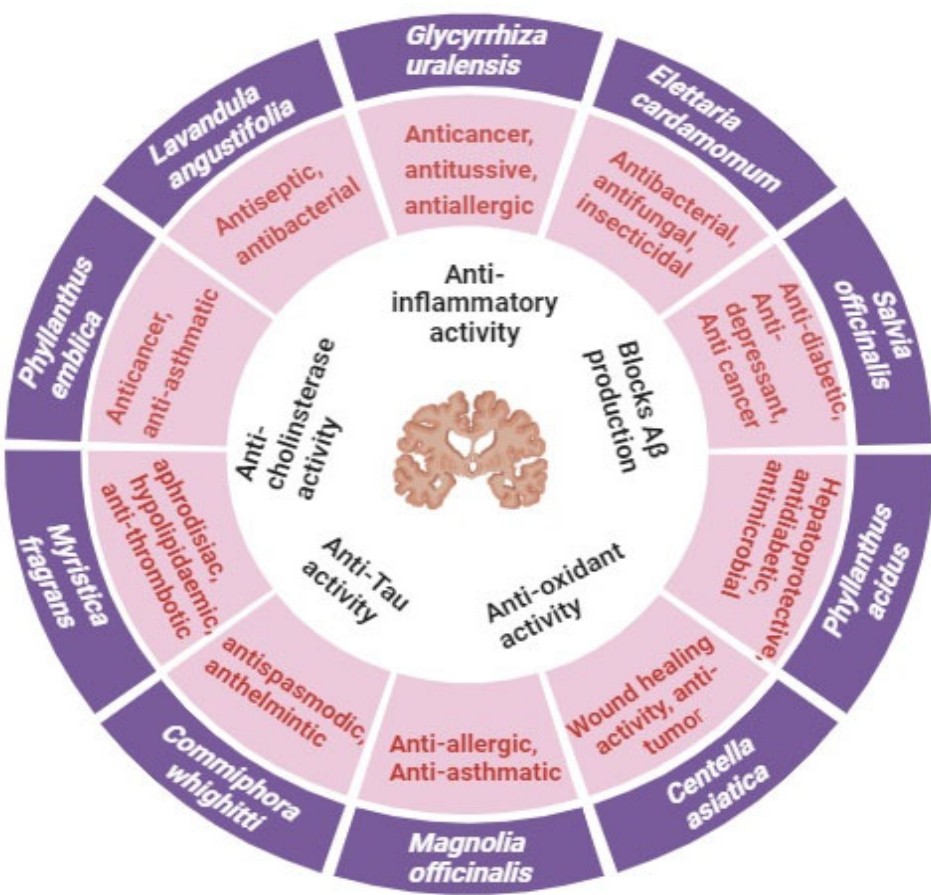

**Figure 3.** Different herbs and their biological activity against AD.

### 4.1. Panax ginseng

PG is an extensively used traditional Chinese medicine belonging to the family Araliaceae and is used to treat a wide range of ailments, including inflammation, cancer, cardiovascular disease, and neurodegenerative disorders [62]. Additionally, in a recent study, it was suggested that due to its neuroprotective properties, it has the potential to cure several diseases associated with the central nervous system, including AD. *Ginseng* contains phytochemicals like saponins, phytosterol, carbohydrates, amino acids, and sugars [63,64]. However, ginsenosides and gintonin are the chief anti-Alzheimer's constituents [65]. Ginsenosides prevent Aβ from aggregation, promote the secretion of neurotrophic factors, and ameliorate mitochondrial dysfunction. Further, ginsenosides have substantial AChE and butyrylcholinesterase (BChE) inhibitory activities, which make these constituents an efficient moiety for reducing the symptoms of AD [15].

In the case of AD, the ACh levels are very low, which is attributed to the elevated levels of AChE and BChE enzymes. AD treatment gains additional advantages from the inhibition of AChE and BChE. In this regard, Choi et al. suggested that according to molecular docking and in vitro tests, the ginsenosides Rb1, Rb2, Rc, Re, Rg1, and Rg3 significantly decrease the activity of AChE and BChE. Moreover, from a group of ginsenosides, the ginsenoside Re seems to possess the optimum AChE inhibitory activity. The inhibitory potential was in the following order: for AChE, Re > Rg3 > Rg1 > Rb1 > Rb2 > Rc, and for BChE, Rg3 > Rg1 > Rb2 > Rb1 > Re > Rc [66]. Throughout the development and progression of AD, there is an elevation of $Ca^{2+}$ levels in the cytosol of neuronal cells. This increase

occurs as a result of $Ca^{2+}$ being transported from both the extracellular space and intracellular stores through mechanisms dependent on transporters. The accumulated $Ca^{2+}$ in neuronal cells possesses the capability to initiate the production and deposition of Aβ plaques and hyperphosphorylated tau associated NFTs. This process contributes to a decline in learning ability observed in individuals with AD. Additionally, transporters located in various cellular structures, such as the cell membrane, endoplasmic reticulum, mitochondria, and lysosomal membranes, play a crucial role in mediating the impact of $Ca^{2+}$ on synaptic plasticity. These transporters are integral to the cognitive decline associated with AD [67]. The signaling pathway known as phosphatidylinositol 3-kinases/protein kinase B (PI3K/AKT) plays a crucial role in signal transduction and controls various biological processes, including cell proliferation, apoptosis, and metabolism. According to available reports, this pathway governs neurotoxicity and facilitates the survival of neurons by acting on different substrates, such as forkhead box protein Os (FoxOs), glycogen synthase kinase-3β (GSK-3β), and caspase-9 [68]. Additionally, it has been found that ginsenoside lowers β-secretase activity, reduces calcium influx, and exhibits a regulatory influence on the PI3K/AKT pathway, thereby lowering the tau protein's phosphorylation. In this context, to clarify the mechanism behind the neuroprotective effects of ginseng, Li et al. investigated the therapeutic effect of ginseng protein (GP) on AD and its correlation with the PI3K/Akt signaling pathway. It was observed that treatment with GP improves the cognitive ability of the AD-affected rats and decreases the level of Aβ. Moreover, in the hippocampus, GP reduced the amount of Aβ$_{(1–42)}$ and pTau and raised the mRNA and protein expression of PI3K, p-Akt/Akt, and Bcl-2/Bax [69]. In addition to ginsenosides, gintonin, a bioactive compound obtained from the glycoprotein fraction of ginseng extract, has also been explored for treating AD. Gintonin has been demonstrated to activate PI3K and AKT, decrease Aβ synthesis in a dose-dependent manner, stimulate the formation of sAPP by activating ADAM10, and not only improve memory impairment brought on by Aβ neurotoxicity but also lessen the deposition of amyloid plaques in a mouse model of AD [70–72]. Moreover, gintonin has been found to improve learning and memory and decrease the production of Aβ by activating the phosphatidic acid receptor, which is implicated in hemolysis [73,74]. Furthermore, gintonin has been reported to exhibit in vitro and in vivo functions against AD through lysophosphatidic acid 1/3 receptors (LPA1/3). Additionally, gintonin can reduce AD symptoms and progression through autophagy stimulation, neurogenesis, decreasing oxidative stress, and anti-inflammatory activities [72].

Recently, Choi et al. demonstrated that systemic treatment with gintonin enhances paracellular permeability of the blood–brain barrier through the LPA1/3 receptor. They also showed that gintonin could improve the brain delivery of donepezil, a representative cognition-improving drug used in AD clinics [72].

In another study, the frontal cortex of mice treated with vehicle, vehicle-treated 5XFAD mice, and non-saponin fraction with rich polysaccharide (NFP)-treated 5XFAD mice were subjected to proteomic analysis to determine the effect of NFP from Korean red ginseng (KRG) on AD. Moreover, 2636 proteins were chosen for hierarchical clustering analysis, and a further 111 proteins were finally identified for protein–protein interaction network analysis. Conclusively, these investigations showed that proteins related to mitochondria and synapses might be associated with the NFP. In addition, subsequent metabolic network analysis also suggested that the therapeutic effects of NFP on AD were correlated with synaptic and mitochondrial-related pathways [75].

### 4.2. Curcuma longa

CL, belonging to the Zingiberaceae family, has been employed to manage health conditions since ancient times due to its medicinal applications [76]. CL, popularly known as turmeric, has long been used due to its healing properties against many diseases, including liver obstruction, jaundice, ulcers, colds, inflammation, and brain disorders. CL con-

tains chemical constituents such as terpenoids, flavonoids, phenylpropene derivatives, alkaloids, and steroids, which exhibit therapeutic effects against the above-stated disorders [77–79]. The inhibitory effect of curcuminoids on acetylcholinesterase activity in AD has already been established by Ahmed and group [80]. According to recent findings, curcumin may offer therapeutic efficacy in the pathogenesis of AD, and in vitro studies suggested that curcumin inhibits Aβ aggregation and Aβ-induced neuroinflammation [81]. Additionally, curcumin has been shown to reduce tau phosphorylation, Aβ deposition, and Aβ oligomerization in vivo, alleviating behavioral impairment in AD animal models [82–84]. Amyloid-oligomers (AβO) are the primary neurotoxin in AD that causes oxidative stress and neuronal degeneration. In a recent study, the protective and anti-oxidative properties of curcumin in AD were assessed. Haibo Yu et al. created an oxidative stress model to simulate an AD cell model using the neuroblastoma SH-SY5Y cell line exposed to AβO. Moreover, the viability of cultured SH-SY5Y cells was considerably decreased after exposure to AβO. In addition, oxidative stress and tau hyperphosphorylation levels were also raised. Furthermore, according to the findings, curcumin reduced ROS production, alleviated oxidative stress, prevented tau hyperphosphorylation, and protected SH-SY5Y cells from AβO damage. Conclusively, curcumin was considered a promising therapeutic agent for managing AD [85].

Curcumin has also been successfully utilized as an anti-inflammatory agent to reduce inflammation in AD [86]. In AD, the deregulation of cyclin-dependent kinase 5 (Cdk5) activity by the production of its hyperactivator p25, leading to the formation of tau and amyloid protein, is an important pathological feature of AD. In this context, Sundaram et al. have revealed an association between p25/Cdk5 hyperactivation and robust neuroinflammation. They confirmed the anti-inflammatory action of curcumin on p25-mediated neuroinflammation and the progression of neurodegeneration in p25Tg mice. Moreover, they demonstrated that curcumin substantially inhibited p25-mediated glial activation and the production of pro-inflammatory chemokines and cytokines in p25Tg mice. Additionally, this curcumin-mediated inhibition of neuroinflammation slowed the advancement of the tau/amyloid pathology caused by p25, which in turn improved the cognitive deficits caused by p25. Conclusively, they suggested that early intervention of inflammation by curcumin could reduce the progression of AD-like pathological outcomes [87]. In addition to pure curcumin, curcumin derivatives have also been reported to exhibit excellent potential against AD. Recently, Utomo and group synthesized curcumin derivatives, i.e., B and N, to act as disaggregation agents against Aβ fibrils. Both the curcumin derivatives, B and N, inhibited fibril formation and induced amyloid disaggregation at low concentrations, thus alleviating Aβ fibril-induced toxicity in neuroblastoma cells. Moreover, both compounds significantly reduced locomotion dysfunction in an Aβ-expressing Drosophila model of AD [88].

In another research, α and β anomers of curcumin glucosides (CGs) were synthesized by Ahmadi and group, using a fusion reaction to examine its in vivo efficacy against AD. The results suggested that both anomers significantly raised levels of glutathione and acetylcholine while significantly lowering lipid peroxidation levels and protein carbonyl levels in brain tissue [89]. Besides the above-discussed findings, CL has also been used with other medicinal plants to treat AD effectively. Recently, CL, in combination with *Citrus junos*, showed an ameliorative effect against Aβ peptide-induced neurological damage. The combined extract was reported to reverse the abnormal behavior and memory impairment caused by Aβ peptide, leading to the recovery of the cholinergic system and ROS oxidative damage defense system [77].

### 4.3. Ginkgo biloba

GB, belonging to the family Ginkgoaceae, is a widely used herbal plant with high medicinal value [90]. The leaves of GB are rich in bioactive compounds like flavonoids, terpenoids, proanthocyanidins, phytosterols, and carotenoids [91,92]. Among them, flavonoids such as biflavones, such as bilobetin, ginkgetin, isoginkgetin, and flavonols, like

quercetin, kaempferol, myricetin, and isorhamnetin, are the chief constituents having medicinal value. Additionally, the terpenoids (ginkgolides A, B, C, J, M, K, and L), sesquiterpene (bilobalide), phytosterols (stigmasterol, $\beta$-sitosterol), and carotenoids ($\alpha$-carotene, $\gamma$-carotene, and lutein) also exhibit marked pharmacological activity. The bioactive compounds obtained from the leaves, exocarp, and seeds of GB are used to treat various diseases such as atherosclerosis, diabetes, asthma, cancer, and AD [93,94]. However, the flavonoids exhibiting antioxidant activity are mainly involved in the management of AD [95]. Natural bioflavones have also been reported to show neuroprotective activities in various models of neurodegenerative disorders through different neuro-pharmacological pathways. The researchers also suggested that GB prevents A$\beta$-induced neuroinflammation and reduces neuroinflammation by preventing the activation of the NLRP3 inflammasome in addition to stimulation of microglial M2 polarization. Thus, GB protects from clinical manifestations of AD [96]. Recently, in a research conducted by Tian et al., ginkgetin, one of the bioflavones, has shown neuroprotective activity by inhibiting neuroinflammatory and apoptotic pathways [97]. Furthermore, in a recent work, a bioactive compound, ginkgolide B (GB), obtained from GB exhibited significant anti-Alzheimer's activity [96]. Additionally, bilobalide, a terpenoid from GB, has been explored to elucidate its structure–activity relationship (SAR) activities by modifying the lactone moiety and simultaneously estimating its neuroprotective effects against A$\beta_{(1-40)}$ peptides and neurite outgrowth activity in PC12 neuronal cells. The derivatives with lactone groups appeared to have pharmacological activity comparable to native bilobalide, whereas those without lactone moieties appeared to have less impact on neurite outgrowth. Therefore, the findings indicated that bilobalide lactone moieties were essential in causing neurite outgrowth effects in PC12 cells [98]. In addition to pure bioactive constituents and their derivatives, crude extract of GB has also displayed anti-AD potential. Recently, the leaf extract of GB has been demonstrated to augment the cognitive behaviors of AD mice, thereby suggesting its therapeutic benefits for AD patients [93]. Xu Liu et al. revealed that long-term oral administration of GB extract EGb 761 alleviated AD pathogenesis by promoting autophagy and reducing neuroinflammation. Moreover, EGb 761 minimized synaptic loss and cognitive impairment in AD, possibly by preventing A$\beta$ aggregation and fostering neuroprotection and neurogenesis [99].

The GB seed was also evaluated for its anticholinesterase activity, and the outcome revealed remarkable changes between the treatment and control groups ($p < 0.001$). One of the main secondary metabolites, vitaxin (identified and confirmed by HPLC), was reported to significantly enhance the anticholinesterase activity of the seed extract. In a nutshell, GB seed was suggested to possess high anticholinesterase properties and low toxicity levels. Thus, the finding justifies the usage of seed as a natural remedy or supplement for neurodegenerative disorders such as AD [100]. Moreover, the extract of leaves of GB belonging to the family Ginkgoaceae exhibits anti-Alzheimer's action by attenuating neurotoxic damages, and this plant also shows antioxidant, inflammatory, apoptotic, and genotoxic effects [93,101].

### 4.4. Centella asiatica

CA, often known as Mandukaparni, is a well-known medicinal plant belonging to the family Umbelliferae; it is used due to its remarkable pharmacological effects for the treatment of various diseases [102]. Extensive research has been conducted to determine the chemical constituents and composition of the plant, and it was found that the plant leaves are rich in carotenoids, terpenoids, phenols, carbohydrates, and amino acids. Moreover, the other phytoconstituents of CA also include tannins, sugars, resins, an alkaloid (hydrocotylin), glycosides (brahminoside, centelloside, asiaticoside A, glycoside D, and asiaticoside B), and phenolic compounds (kaempferol, chlorogenic acid, quercitrin, and luteolin). In addition, CA contains centellosides (pentacyclic triterpenoid saponins), which are considered to be one of the main bioactive compounds of the plant [103]. CA has captivated interest as a potent medicinal plant with useful pharmacological activities,

including antifungal, antidiabetic, antioxidant, antiulcer, cardioprotective, immunostimulant, and hepatoprotective activities. Additionally, the plant has long been used as a neuroprotective herb for the management of various CNS disorders such as epilepsy, dementia, AD, and anxiety. Among these, the CA extract has been shown to exhibit promising improvement in learning and memory in AD patients [104,105]. Research reports have proven CA to be a potential anti-AD herbal drug. In a preclinical animal model, CA was reported to attenuate hippocampal mitochondrial dysfunction and improve memory and executive function in β-amyloid-overexpressing mice [106]. The herbal drug has also been reported to attenuate Aβ-induced neurodegenerative spine loss and dendritic simplification, leading to improvement in symptoms of AD [107].

In a recent work, CA extract exhibited neuroprotective effects in rats with damaged nucleus basalis of Meynert (NBM). The results indicated that rats with impaired NBM had enhanced cognitive abilities when CA was administered, regardless of dose. This was accomplished because CA reduced the acetylcholinesterase level while restoring cholinergic dysfunction. The findings also suggested that memory could be improved by administering CA extract for six weeks prior to inducing NBM lesions and AD induction [108]. Witter and coworkers also proved the anti-AD potential of CA. They examined CA and seven other plants for inhibition of amyloid-$\beta_{(1-40)}$ (A$\beta_{40}$) and methionine amyloid-$\beta_{(1-40)}$ (MA$\beta_{40}$) fibrillation, which causes the formation of plaque leading to dementia and AD. It was observed that the asiaticosides present in CA extract act as the most potent inhibitor, which was found to interact with MAb40 [109]. In this sequence, Chen and coworkers investigated the anti-Alzheimer's activity of CA extract and revealed the ability of CA extract to inhibit Aβ-mediated reactive oxygen species production in neural cells [110]. The potential of CA extract to increase cytochrome B, NADH dehydrogenase 1, cytochrome C oxidase 1, and ATP synthase 6 in the hippocampal neurons, which expand on the cognitive effect, was further reported by Matthews and group [107]. Apart from the crude extract, pure phytoactives of CA have also been explored for the management of AD. In a recent work, a bioactive compound, asiaticoside, a natural glycoside obtained from CA leaves, was investigated for its anti-AD effect. In this context, Song and group examined the protective effect of asiaticoside on A$\beta_{(1-42)}$-induced cytotoxicity, apoptosis, and related mechanisms in human brain microvascular endothelial cells (hBMECs). The findings demonstrated that pre-treatment with asiaticoside (25, 50, and 100 M) for 12 h dramatically reduced the inhibition of cell growth and apoptosis and restored the decreased mitochondrial membrane potential produced by A$\beta_{1-42}$ (50μM) in hBMECs. Additionally, in hBMECs, asiaticoside dramatically reduced the raised expressions of TNF-, IL-6, TLR4, MyD88, TRAF6, and p-NF-κB p65. It also prevented the concentration-dependent translocation of NF-κB p65 from the cytoplasm to the nucleus that was brought about by A$\beta_{(1-42)}$. Furthermore, it was suggested that the possible molecular mechanism of asiaticoside may involve blocking the TLR4/NF-B signaling pathway, and asiaticoside might be regarded as a novel therapeutic compound for the prevention and treatment of AD [111]. In addition, previous studies have shown that asiatic acid (AA) (major bioactive triterpene) may be helpful in reducing Aβ levels in the AD brain with subsequent reduction in the pathologies associated with AD [112]. Cheng et al. showed that AA shields differentiated PC12 cells from apoptosis and tau protein hyperphosphorylation (pTau) caused by A$\beta_{25-35}$, which may be partially mediated via activating the PI3K/Akt/GSK-3β signaling pathway. The overall findings suggested that AA may be potentially used to treat AD [113].

The extract of the leaves of *Centella asiatica* belonging to the family Umbelliferae exhibits anti-Alzheimer's action by ameliorating cognitive impairment. The main active constituents of this plant includes triterpenoid saponin, asiaticoside, madecassoside, asiatic acid, and madecassic acid. In addition to anti-AD action, the plant also displays other biological activities like antioxidant properties and antitumor, neuroprotective, and wound-healing activity [108,114].

### 4.5. Glycyrrhiza Species

*Glycyrrhiza*, one of the most widely used herbs in the world, belonging to the Leguminosae family, has 29 species and six varieties worldwide [115]. The three species that are considered medicinal *Glycyrrhiza* plants in Chinese Pharmacopoeia include *Glycyrrhiza glabra* (GB), *Glycyrrhiza uralensis* (GU), and *Glycyrrhiza inflate* (GI). The various pharmacological activities of *Glycyrrhiza* are due to its wide range of chemical components, including flavonoids (liquiritin, liquiritigenin, chalcones isoliquiritin, and isoliquiritigenin), isoflavonoids (glabridin, galbrene, and shinpterocarpin), stilbenoids, coumarins, alkaloids, polysaccharides, triterpenoid saponins, and proteins. The dried roots and rhizomes of the plant have been used clinically for centuries due to their expectorant, antiulcer, anxiolytic, diuretic antipyretic, and antimicrobial activities [116]. Moreover, GB has been found to have antioxidant, anti-inflammatory, and neuroprotective effects in the management of neurodegenerative disorders related to dementia [117,118]. *Glycyrrhiza* crude extract, active phytoconstituents, and their derivatives have been reported to exhibit an excellent anti-AD potential by acting on different events of AD, for instance, inhibition of pTau and Aβ inhibition. Recently, liquiritigenin, an important compound found in GB (licorice), was screened for its inhibitory action against tau amyloid formation. The researchers revealed that liquiritigenin could stop the formation of tau amyloid fibrils in a way that exhibits fewer cytotoxic effects. Conclusively, they suggested that liquiritigenin could be employed as a promising inhibitor of amyloid fibril formation in the treatment of AD [119].

Glycyrrhizin, a triterpenoidal saponin with isoliquiritigenin as a phenolic component, reduces oxidative stress and degeneration of brain cells in AD and dementia patients [117]. Furthermore, beta-site amyloid precursor protein-cleaving enzyme 1 (BACE1), a significant molecule involved in the etiology of AD, has been reported to be inhibited by the phytoactive compounds present in Glycyrrhiza species. Recent research conducted by Nadh et al. showed that out of the 11 compounds screened, one of the bioactive compounds, N-(4-hydroxybutyl) phthalimide obtained from GB presented an inhibitory effect against BACE1 as confirmed by structure-based docking approach, and, conclusively, it was suggested that N-(4-hydroxybutyl) phthalimide could act as a potential plant-derived BACE1 inhibitor for the treatment of AD [71]. In a recent work, a bioactive compound, glycyrrhizic acid, obtained from GU, was investigated for its neuroprotective role in the scopolamine-induced memory impairment mice model. In this regard, recently, Ban et al. observed that glycyrrhizic acid significantly reduced the cognitive impairment caused by scopolamine. Moreover, when compared to the scopolamine-treated group, the superoxide dismutase activity, AChE activity, and catalase activity in the glycyrrhizic acid-treated group significantly reversed cognitive impairment may be due to the increased activity of those enzymes. As a result, the findings imply that glycyrrhizic acid had a neuroprotective role in cognitive impairment induced by scopolamine [120]. Furthermore, licochalcone B (LCB) obtained from the root of GI was investigated for its anti-AD activity, and the result revealed that LCB could prevent Aβ$_{42}$ aggregation by preventing salt bridge interaction at the C-terminus and chelating metal ions. Moreover, in SH-SY5Y cells, LCB also displayed neuroprotective and antioxidant effects. Therefore, the findings imply that LCB is a promising, multifunctional drug with minimal toxicity and significant efficacy for the management of AD [121].

The extract of the leaves of GU, belonging to the family Leguminosae, exhibits anti-AD action by showing protective action against glutamate-induced mitochondrial damage and hippocampal neuronal cell death. The main active constituents of this plant include isoliquiritigenin, flavonoids, and saponins. In addition to anti-AD action, the plant also displays other biological activities such as antitussive, antiallergic, anticancer, and anti-inflammatory activity [122].

### 4.6. Bacopa monnieri

BM, belonging to the family Plantaginaceae, often known as the miracle tree or the tree of life, is widely utilized as a functional food and nutritional supplement worldwide [123]. BM, popularly known as Brahmi, is a nootropic ayurvedic plant that has been used for centuries to treat various neurological diseases. In fact, this medicinal herb possesses many pharmacological activities such as antioxidant, antibacterial, antifungal, antidiabetic, antihypertensive, antitumor, hepatoprotective, antineoplastic, bronchodilatory, and immunostimulatory effects [124]. Additionally, the plant is rich in bioactive compounds such as saponins: bacoside-A (bacopaside V, bacopaside II, bacopaside IX, bacopasaponin A-H), glycosides (asiaticoside and thanakunicide), alkaloids (herpestine and brahmine), sapogenin (jujubacogenin), and various phytochemicals, such as stigmasterol, wogonin, and β-sitosterol [125]. Moreover, several neuroprotective triterpenoidal saponins such as bacoside-A, bacoside-B, bacosaponins, and betulinic acid play a major role in ameliorating memory and cognitive functions [124,126]. Interestingly, the research findings support the use of an extract of BM for the treatment of AD, as it shows similarities in the cholinergic effects comparable to those of donepezil, rivastigmine, and galantamine [125]. Another research finding demonstrates that BM inhibits cholinergic degradation and enhances cognition in a rat AD model [15]. Bacosides, a group of compounds derived from the BM plant, demonstrate noteworthy therapeutic characteristics, particularly in enhancing cognitive functions and potentially exhibiting anti-amyloid activity. Our findings indicate that bacoside-A has notable impacts on the fibrillation process and membrane interactions of the amyloidogenic fragment of the prion protein (PrP(106–126)). Specifically, when co-incubated with PrP(106–126), bacoside-A expedites the formation of fibrils in the presence of lipid bilayers while simultaneously impeding the interactions of the peptide aggregates with the bilayers formed in solution. Additionally, spectroscopic and microscopic methods were used to investigate these intriguing events, and the results of the study imply that the underlying mechanisms by which bacoside-A reduces the toxicity of amyloid protein may involve the induction of fibril formation by reducing the concentration of membrane-active pre-fibrillar species of the prion fragment and corresponding suppression of membrane interactions [127].

BM leaves are well established for improving memory and are much sought after for treating neurodegenerative diseases. In this context, in recent research, BM shows significant results by reducing oxidative and mitochondrial stress and enhancing life span while reducing aging symptoms in *Caenorhabditis elegans*. The results demonstrate that in HT-22 cells, the hexane extract of BM (but not an ethanol extract) reduced the toxicity of 5 mM glutamate. The mechanism behind this includes lowering ROS generation and preventing ER and mitochondrial stress [128].

Apart from the above studies, bacoside-A3 suppressed the production of oxidative radicals, PGE2, and the production of iNOS, thus preventing the amyloid-mediated decrease in U87MG cell viability. Additionally, in U87MG cells, bacoside-A3 prevented the NF-κB from entering the nucleus. Therefore, further in vivo research is required since bacoside-A3 offers therapeutic potential for AD [129].

### 4.7. Tinospora cordifolia

TC, also known as Guduchi, which belongs to the family Menispermaceae, is a perennial climbing herb that is widely distributed throughout the world [130]. TC is a conventional medicine with widespread pharmacological activity such as immunomodulatory, antidiabetic, antihypertensive, hepatoprotective, anti-inflammatory, anticancer, antipyretic, and cardioprotective effects. The biologically active chemical compounds found in TC are among the several chemical classes of compounds, including glycosides, terpenoids, alkaloids, steroids, flavonoids, diterpenoid lactones, and phenolic acids [131]. The multiple conventional medical applications of the plant have drawn the attention of researchers from all around the world for more than half a century. One study suggested

that the metabolites isolated from TC could potentially be used for the treatment of AD. Tinosporide and 8-hydroxytinosporide, obtained from a methanolic extract of TC, have been investigated for their inhibitory efficacy against AChE and BuChE. The tinosporide shows significant anti-AChE activity. The results of the docking study indicated that donepezil's complementary novel ligand, tinosporide, would be a suitable lead for the development of drugs to be utilized in the treatment of cognitive impairment, as tinosporide is a powerful inhibitor of the AChE enzyme.

The findings suggested that tinosporide would be a complementary noble molecule of donepezil, which is correlated with its pharmacological activity through in vitro studies, whereas 8-hydroxytinosporide inhibited BuChE modestly, and the results are very close to the standard donepezil [132]. Another finding suggested that supplementing TC to a person suffering from behavioral issues and memory impairment may improve thinking skills and memory [133]. Several clinical trials have provided evidence that ginkgo enhances memory, reduces memory loss, improves concentration, and alleviates anxiety in individuals with dementia [134]. The antioxidant and anti-inflammatory properties of TC leaves may be due to their water and alcoholic extracts. In this context, the results of a recent finding suggested that TC dry leaf extracts showed anti-inflammatory and antioxidant properties through the upregulation of antioxidant enzymes and attenuation of NF-κB nuclear translocation in activated human monocytic (THP-1) cells. The study, therefore, supports the proposed molecular basis for the traditional use of TC for treating various inflammatory diseases such as AD [135]. One of the recent exciting studies showed that the extract of *Nigrospora oryzae*, an endophytic fungus isolated from TC, is tested for AChE inhibition and anticholinesterase activity. The extract showed modulation of the cholinergic pathway in the scopolamine-induced model of dementia, which resulted in AChE inhibition and antioxidant activity. Also, the findings suggested that the anti-dementia-like activity is due to the potential bioactive compound quercetin produced by *Nigrospora oryzae*, which performs a crucial role in the management of AD [136].

*4.8. Convolvulus pluricaulis*

CP, commonly referred to as Shankhapushpi, belonging to the family Convolvulaceae, is among the best nerve tonic for treating nervous disorders [137]. Shankhapushpi exhibits a wide range of neuroprotective effects, including mental stimulant, anxiolytic, tranquilizing, immunomodulatory, anti-depressant, neurodegenerative, anti-convulsant, anti-inflammatory, antioxidant, and anti-AD effects [138]. In addition, the most effective bioactive constituents in CP, which exhibits neuroprotective activity, include cinnamic acid, linoleic acid, β-sitosterol, pentanoic acid, vitamin E, phthalic acid, ascorbic acid, tropane alkaloids, and kaempferol [139]. Interestingly, the currently approved drugs for treating AD in modern medicine only have minor, temporary effects in ameliorating the disease manifestation and are barely able to stop or reverse the condition. Therefore, the study was conducted to assess how Shankhapushpi feeding affects the *Drosophila melanogaster* (common fruit fly) model of AD, and the findings suggested that prophylactic consumption of Shankhapushpi reduced the neurodegenerative symptoms seen in the *Drosophila melanogaster* AD model [140].

Indian traditional medicine exploits the aerial parts of the CP to treat neurological disorders. The study uncovers the effects of CP extract (CPE) on a chronic rat model of depression, and the findings revealed that in the (chronic unpredictable mild stress) CUMS-exposed rats, fluoxetine (10 mg/kg, p.o.) or CPE (50, and 100 mg/kg) treatments given back-to-back over one week markedly increased sucrose preference index, decreased immobility time in the FST, and increased the number of squares crossed, and locomotion in the actophotometer.

CPE (50 and 100 mg/kg) or fluoxetine administration significantly decreased the increased levels of pro-inflammatory cytokines IL-1, IL-6, TNF-, and liver biomarkers ALT and AST in the CUMS-exposed rats. As a result, this study showed that CPE had an anti-

depressant-like impact on stressed rats, restoring liver biomarkers or monoaminergic responses, which might be mediated via anti-inflammatory potential [141]. CP has been reported to show antioxidant potential by its capacity to scavenge free radicals. Therefore, the plant might be speculated as a plant-based anti-Alzheimer's agent. In this regard, Rachita and the group reported that pre-treatment of CP reduces oxidative stress and reduces apoptosis caused by the $H_2O_2$ that causes neuronal damage. These findings imply that CP may be used to treat neurodegeneration caused by stress [139].

### 4.9. Withania somnifera

WS, belonging to the family Solanaceae, is a plant that has had a significant medicinal value in Ayurveda for a long time [142]. The traditional uses of this plant suggest that it has positive implications against a plethora of human diseases, such as cancer, diabetes, asthma, stress, and hypertension [143]. The different classes of phytochemicals involved in WS include alkaloids (withanine, somniferine, and somnine), flavonoids (kaempferol, quercetin), steroidal lactones (withaferin-A, withanolides, and withanone), steroids (stigmasterol, sitoinosides, and stigmastadien), and N-containing compounds, such as somnisol, withnanol, etc. [144]. The bioactive compounds and extracts derived from WS have been reported to exhibit potent pharmacological activity, such as antimicrobial, antidiabetic, anti-inflammatory, hepatoprotective, cardioprotective, hypoglycemic, and immunomodulatory activity. Moreover, it was found that WS is predominantly active against many neurological conditions like AD, Parkinson's disease, and Huntington's disease [144,145]. Furthermore, the plant contains the active ingredient withanone, which has extraordinary pharmacological and neurological effects [146]. Ashwagandha has an ancient legacy of application in traditional medicine to improve memory and cognitive function. In this regard, a pilot study was conducted by Choudhary and the group to determine whether *Ashwagandha* is safe and effective at improving memory and cognitive performance in adults with mild cognitive impairment (MCI). The results suggested that *Ashwagandha* improves memory in a person with MCI and enhances attention, executive function, and information processing speed [147].

According to the simulation studies, it was investigated whether withanamides A and C may bind to the active motif of ($A\beta_{25-35}$) distinctly and may shield cells from the toxicity caused by $A\beta$ by preventing fibril formation [148]. Furthermore, the derivatives of WS (withanolide A, withanolide B, withanoside IV, and withanoside V) have been studied for their ameliorative effects on the amyloid-$\beta_{42}$ fibril formation for AD. The cytotoxic activity tested on the human SK N-SH cell line reported IC 50 value to be 28.61 ± 2.91, 14.84 ± 1.45, 18.76 ± 0.76, and 30.14 ± 2.59 μM, respectively. In addition, the tunnel assay showed reduced DNA damage and a significant decrease in apoptotic cells following treatment with WS derivatives. Additionally, molecular docking findings show that the derivatives interacted with the hydrophobic core residues area of -$\beta$ amyloid, preventing the binding of another -$\beta$ amyloid peptide and, therefore, preventing fibrillization. Moreover, it appears from 100 ns simulations that the withanolide B and withanoside V complexes were stable inside the hydrophobic core of $\beta$-amyloid [149].

### 4.10. Celastrus paniculatus

*Celastrus paniculatus* (CP), a member of the Celastraceae family, has demonstrated noteworthy neuroprotective attributes attributed to its antioxidant activity[150]. The administration of CP has been shown to mitigate neuronal cell damage caused by hydrogen peroxide and glutamine-induced toxicity[151]. Notably, the botanical extract of CP has been observed to enhance cholinergic activity, thereby contributing to its efficacy in improving memory performance [152]. The aqueous extract of CP exhibits both antioxidant and cognition-enhancing properties [153]. Specifically, it safeguards neuronal cells against hydrogen peroxide-induced toxicity, underscoring its potential therapeutic role. This neuroprotective effect is, in part, attributed to the extract's ability to act as an antioxidant and free radical scavenger. The findings highlight the promising role of CP in neuroprotection

and cognitive enhancement, offering potential avenues for further exploration and development of novel therapeutic interventions [154].

Apart from the above-discussed plants, numerous plants that have been successfully employed for the treatment of AD have been illustrated in Table 1.

**Table 1.** Traditional Indian medicinal plants beneficial in Alzheimer's disease.

| S. No. | Plant/Herb | Family | Part Studied | Chemical Active Compounds | Mode of Anti-Alzheimer's Action | Reference |
|---|---|---|---|---|---|---|
| 1 | *Citrus limon* | Rutaceae | Fresh lemon juice | Flavonoids, vitamin C, poly phenol, folic acid, potassium, pectin | Improve the cognitive performance, *Citrus limon* juice improved cholinergic neurotransmission and enhanced the antioxidant system | [122] |
| 2 | *Elettaria cardamomum* | Zingiberaceae | Ethanolic extract of *E. cardamomum* (seeds) | Alpha-terpinyl acetate, phenolic compounds, flavonoids, and tannins | AChE enzyme inhibition, BuChE enzyme inhibition, decrease Aβ-induced neurotoxicity, reduced oxidative stress induced by hydrogen peroxide, antioxidant activity, and anti-amyloidogenic activity | [155,156] |
| 3 | *Salvia officinalis* | Lamiaceae | Essential oil, ethanolic extract | Flavonoids, terpenoids, and essential oil | AChE inhibitory activity and pathogenesis of dementia | [157] |
| 4 | *Phyllanthus acidus* | Phyllanthaceae | Ethanolic extract of leaves | Triterpene, diterpene, sesquiterpene, and glycosides | The ethanolic extract significantly decreased lipid peroxidase and increased super oxidase dismutase and brain catalase activities against arsenic-induced neurotoxicity, AChE inhibitory potential | [158,159] |
| 5 | *Pistacia vera* | Anacardiaceae | Fruit extract | Flavonoids, phenolic, essential oil | Ameliorates cognitive process in scopolamine-induced Swiss albino mice, anti-Aβ aggregation, anti-neuroinflammatory properties, and AChE inhibitory activity | [15,160] |
| 6 | *Lepidium meyenii* | Brassicaceae | Dried hypocotyls aqueous and hydroalcoholic extracts | Polysaccharides, alkaloids, and polyphenols | Ameliorates the scopolamine-induced memory deficit, inhibits AChE activity | [161] |
| 7 | *Magnolia officinalis* | Magnoliaceae | Extract of stem bark | Honokiol derivatives, meroterpenoids, lignans, glycosides, alkaloids | AChE and BChE inhibitory activity | [162,163] |
| 8 | *Commiphora whighitti* | Burseraceae | Resin | Guggulsterone, guggulipid | AChE inhibition | [164,165] |
| 9 | *Celastrus paniculatus* | Celastraceae | Seed oil | Triterpenoids and sesquiterpenes | Improves ACh level | [164,166] |
| 10 | *Myristica fragrans* | Myristicaceae | Seed | Myristicin, elemicin, safrole, myristic acid, alpha-pinene | Improves memory deficit | [164,167] |
| 11 | *Hibiscus rosa-sinensis* | Malvaceae | Buds and flowers ethanolic extract | Flavonoids, glycosides, quercetin 3-O-sophoroside | Reversed the scopolamine-induced decrease in ChAT expression, increased AChE expression, and decreased ACh | [168,169] |

| 12 | *Phyllanthus emblica* | Euphorbia-ceae | Fruit | Polyphenols, myricetin, quercetin, fisetin, and gallic acid | Inhibition effect on AChE, improves memory | [170] |
|----|------|------|------|------|------|------|
| 13 | *Coriandrum sativum* | Apiaceae | MeOH extract of the aerial parts | Glycosides, α-ter-pinene, linalool, a-pinene | Anti-neuroinflammatory activity, potent NGF secretion activity | [171] |
| 14 | *Ficus carica* | Moraceae | Crude extract of mesocarp | γ-sitosterol, umbellifer-one, rutin, anthocya-nin, coumarins | Reduce oxidative stress | [172,173] |
| 15 | *Lavandula angustifolia* | Lamiaceae | Leaves | Linalool, 1,8-cineole, li-nalyl acetate, lavan-dulyl acetate | Improves memory deficit | [174] |

## 5. Clinical Trials

The widespread use of plant-based medicines does not assure the safety and efficacy of these medicines. Plant-based medicines have many chemical constituents with complex pharmacological effects on the human body. Although herbal practitioners and believers do not require clinical trials, it is necessary for large-scale ethical acceptance and viability in the international market as it is the need of the time. The development of scientific, evidence-based pharmacological and clinical data for plant-based medicinal products can break the hurdle and initiate the integration of herbal medicines into conventional medical practices [175]. Traditional medicine formulation, when developed with characterization and quantification of bioactive or marker phytocompounds, can achieve global regulatory acceptance. There are trials of several promising and potent phyto-based formulations to treat AD, which are currently in the recruiting phase or in different stages. These formulations are anticipated to be potential sources in the development of anti-AD and other medicaments [176–178]. To summarize, while plant-based medicines have gained popularity, clinical trials are required to verify their safety and efficacy. This will allow herbal medicines to be integrated into conventional medical procedures and boost their prospects of receiving regulatory approval on a worldwide basis. Clinical trial data of some promising anti-AD formulations/agents have been presented in Table 2. We used the information source for this review from the FDA/U.S. National Library of Medicine of the National Institutes of Health (NIH) clinical research registry, ClinicalTrials.gov.

**Table 2.** Summary of clinical trial conducted for anti-Alzheimer's medicinal plants.

| NCT Number | Title | Status | Interventions | Phase | Population | Sponsor | Result |
|---|---|---|---|---|---|---|---|
| NCT00391833 | Effect of *Panax ginseng* on the Cognitive Performance in AD | Completed | *Panax ginseng* | Phase 1 Phase 2 | Enrollment: 97 Age: 40 years to 83 years (adult, older adult) | Seoul National University Hospital, Republic of Korea | Enhanced cognitive metrics observed in Alzheimer's patients. Improvement in memory retention and recall demonstrated significant potential for *Panax ginseng* as a therapeutic agent in cognitive decline associated with AD. |
| NCT03221894 | A Retrospective Study to Investigate the Additive Effectiveness of Chinese Herbal Medicine in AD | Completed | Dietary supplement: GRAPE granules | Not available | Enrollment: 120 Age: 50 years to 85 years (adult, older adult) | Dongzhimen Hospital, Beijing, China; Beijing Hospital, China; Chinese PLA General Hospital, China; Peking University Third Hospital, China | Chinese herbal medicine, specifically GRAPE granules, exhibits supplementary efficacy in mitigating cognitive decline in Alzheimer's patients. Notable improvements in memory consolidation and attentional functions were observed. |
| NCT04570644 | Randomized I/II Phase Study of ALZT-OP1 Combination Therapy in AD and Normal Healthy Volunteers | Completed | ALZT-OP1 (cromolyn and ibuprofen); ALZT-OP1a (cromolyn) and ALZT-OP1b (ibuprofen) | Phase 1 | Enrollment: 56 Age: 55 years to 79 years (adult, older adult) | AZ Therapies, Inc., Boston, Massachusetts, USA | ALZT-OP1 combination therapy exhibited promising results in both Alzheimer's patients and healthy volunteers. Reduction in neuroinflammatory markers and cognitive enhancement were observed, indicating a potential breakthrough in disease-modifying interventions. |
| NCT04149860 | Study With Lu AF87908 in Healthy Participants and Participants with AD | Recruiting | Lu AF87908; Placebo | Phase 1 | Enrollment: 88 Age: 18 years to 65 years (adult, older adult) | H. Lundbeck A/S, Copenhagen, Den-mark | Ongoing trial to assess the safety and preliminary efficacy of Lu AF87908 in both healthy individuals and those afflicted with AD. Early indications suggest a potential for cognitive enhancement, warranting further investigation. |

| | | | | | | |
|---|---|---|---|---|---|---|
| NCT02547 818 | Safety and Efficacy Study of ALZT-OP1 in Subjects with Evidence of Early AD | Completed | ALZT-OP1a; ALZT-OP1b; Placebo ALZT-OP1a; Placebo ALZT-OP1b | Phase 3 | Enrollment: 620 Age: 55 years to 79 years (adult, older adult) | AZ Therapies, Inc., Boston, Massachusetts, USA | ALZT-OP1 demonstrates safety and substantial efficacy in subjects with early-stage AD. Robust cognitive preservation and a significant reduction in disease progression were observed, supporting its potential as a disease-modifying therapy. |
| NCT04300 569 | A Study to Determine the Signs and Symptoms that Impact Daily Life of Participants with Irregular Sleep-Wake Rhythm Disorder | Completed | Non-interventional | Not available | Enrollment: 37 Age: 18 years to 90 years (adult, older adult) | Eisai Inc., Tokyo, Japan | The study delineated the significant impact of irregular sleep–wake rhythm disorder on participants' daily lives. Key indicators affecting cognitive function were identified, shedding light on potential therapeutic avenues. |
| NCT05591 027 | Safety and Target Engagement of *Centella asiatica* in Cognitive Impairment | Not yet recruiting | *Centella asiatica* product; Placebo | Phase 1 | Enrollment: 48 Age: 65 years to 85 years (older adult) | Oregon Health and Science University Alzheimer's Association, USA | Anticipated study aims to ascertain safety and efficacy of *Centella asiatica* in cognitive impairment. Preliminary data suggest promising neuroprotective effects, warranting further investigation. |
| NCT05269 173 | Efficacy and Safety of *Flos gossypii* Flavonoids Tablet in the Treatment of AD | Recruiting | *Flos gossypii* flavonoids tablet | Phase 2 | Enrollment: 240 Age: 50 years to 85 years (adult, older adult) | Capital Medical University, China; Xinjiang Uygur Pharmaceutical Co., Ltd., China | Ongoing investigation into the potential of *Flos gossypii* flavonoids tablet in treating AD. Preliminary data suggest a favorable impact on cognitive function and disease progression. |
| NCT03286 608 | Polyphenols and Risk of Dementia | Completed | Observational study (no intervention) | Not available | Enrollment: 1329 Age: 65 years and older (older adult) | Jean-François Dartigues, France; University of Bordeaux, France | Observational study elucidating the relationship between polyphenols and dementia risk. Data underscore a potential protective effect, warranting deeper mechanistic exploration. |
| NCT00205 179 | AD: Potential Benefit of Isoflavones | Completed | Novasoy; Placebo | Phase 2 | Enrollment: 72 Age: 55 years and older (adult, older adult) | University of Wisconsin, Madison, USA; National Institutes of Health (NIH), USA; | Isoflavones, specifically novasoy, exhibit potential benefits in AD. Signifi- |

| | | | | | National Institute on Aging (NIA), USA | cant improvements in cognitive metrics were observed, suggesting a role in disease management. |
|---|---|---|---|---|---|---|
| NCT00500 500 | Effect of EGb 761® on Patients with Mild to Moderate AD | Terminated | EGb 761® (Tanakan®) | Phase 2 | 50 years to 85 years (older adult) | Ipsen, Paris, France | Trial exploring the effect of EGb 761® on patients with mild-to-moderate AD. Study discontinued prematurely, with limited conclusive data available. |
| NCT00164 749 | A Pilot Study of Curcumin and Ginkgo for Treating AD | Completed | Placebo and ginkgo extract; Curcumin and ginkgo extract | Phase 1 Phase 2 | 50 years and older | Chinese University of Hong Kong; BUPA Foundation, Hong Kong; Kwong Wah Hospital, Hong Kong | Pilot study investigating the potential benefits of curcumin and ginkgo in AD. Preliminary data suggest a positive impact on cognitive function, warranting further exploration. |
| NCT00276 510 | A Study of EGb 761® (Tanakan®) in Dementia of Alzheimer's-Type Onset in Patients Suffering From Memory Complaints | Completed | EGb 761® (Tanakan®); Placebo | Phase 4 | 70 years and older | Ipsen, Paris, France | Study assessing the impact of EGb 761® in patients with dementia of Alzheimer's-type onset. Data indicate potential cognitive benefits, suggesting a role in early intervention. |
| NCT00010 803 | *Ginkgo biloba* Prevention Trial in Older Individuals | Completed | *Ginkgo biloba* Placebo | Phase 3 | 75 years and older | National Center for Complementary and Integrative Health (NCCIH), USA; Office of Dietary Supplements (ODS); National Institute of Neurological Disorders and Stroke (NINDS), USA; National Institute on Aging (NIA), USA; National Heart, Lung, and Blood Institute (NHLBI), USA | Comprehensive trial investigating the preventive potential of *Ginkgo biloba* in older individuals. No significant cognitive benefits were observed, necessitating further research into alternative interventions. |

| | | | | | | |
|---|---|---|---|---|---|---|
| NCT03090 516 | Clinical Efficacy of *Ginkgo biloba* Extract in the Treatment of AD | Unknown status | *Ginkgo biloba* dispersible tablets; Donepezil; *Ginkgo biloba* dispersible tablets and donepezil | Phase 2 Phase 3 | 50 years to 85 years | The First Affiliated Hospital with Nanjing Medical University, China | Ongoing investigation into the clinical efficacy of *Ginkgo biloba* extract in AD. Preliminary data are pending, with outcomes yet to be determined. |
| NCT01001 637 | Efficacy and Safety of Curcumin Formulation in AD | Unknown status | Dietary supplement: Curcumin formulation; Dietary supplement: Placebo | Phase 2 | Enrollment: 26 Age: 50 years to 80 years (adult, older adult) | Jaslok Hospital and Research Centre, Mumbai, India; Pharmanza Herbal Pvt Ltd., Gujarat, India; Verdure Sciences; University of California, Los Angeles, USA | Study evaluating the efficacy and safety of curcumin formulation in AD. Preliminary data are pending, with outcomes yet to be determined. |
| NCT00099 710 | Curcumin in Patients with Mild to Moderate AD | Completed | Dietary supplement: Curcumin C3 complex | Phase 2 | Enrollment: 33 Age: 50 years and older (adult, older adult) | John Douglas French Alzheimer's Foundation, USA; Institute for the study of Aging (ISOA), USA; National Institute on Aging (NIA), USA | Investigation into the potential benefits of curcumin C3 complex in patients with mild-to-moderate AD. Preliminary data suggest positive cognitive effects, warranting further exploration. |
| NCT01811 381 | Curcumin and Yoga Therapy for Those at Risk for AD | Unknown status | Curcumin; Behavioral: Aerobic yoga; Behavioral: Non-aerobic yoga; Dietary supplement: Placebo | Phase 2 | Enrollment: 80 Age: 50 years to 90 years (adult, older adult) | VA Office of Research and Development, USA | Ongoing trial assessing the combined impact of curcumin and yoga therapy in individuals at risk for AD. Preliminary data are pending, with outcomes yet to be determined. |

| NCT01716 637 | Short Term Efficacy and Safety of Perispinal Administration of Etanercept in Mild to Moderate AD | Completed | Etanercept; Dietary supplement: Curcumin, Luteol, Theaflavin, Lipoic Acid, Fish Oil, Quercetin, Resveratrol | Phase 1 | Enrollment: 12 Age: 60 years to 85 years (adult, older adult) | Life Extension Foundation Inc., USA | Study investigating the short-term efficacy and safety of perispinal administration of etanercept in mild-to-moderate AD. Preliminary data indicate a potential for cognitive improvement, warranting further exploration. |
|---|---|---|---|---|---|---|---|
| NCT04606 420 | Lifestyle Changes can Reverse Early-Stage AD | Active, not recruiting | Behavioral: Lifestyle medicine | No data available | Enrollment: 51 Age: 45 years to 90 years (adult, older adult) | Preventive Medicine Research Institute; University of California, San Francisco, USA; Harvard Medical School (HMS and HSDM), USA; University of California, SanDiego, USA; The Cleveland Clinic; Renown Health, USA | Ongoing investigation into the potential of lifestyle changes in reversing early-stage AD. Preliminary data are pending, with outcomes yet to be determined. |

## 6. Conclusions and Future Direction

In conclusion, the narrative offered by the authors sheds light on key facets of the neuroprotective function of natural antioxidants in the aging process associated with AD. AD is a neurological condition that worsens over time, and there is no known cure. The unmet need for a comprehensive understanding of the illness has driven global scientific efforts to pave the way for various preventive treatment approaches to AD. AD is now a significant global epidemic. Drugs already on the market merely treat symptoms; they do not stop the onset or spread of disease. It has been noted that the number of phytochemicals with therapeutic activity has increased during the last two to three decades. Recently, developed and developing nations have dramatically increased their usage of plant-based health products, which has expanded the worldwide market for plant goods.

One of the most promising avenues for herbal medicine is that they have a diverse range of medical applications, such as antioxidant activity and inhibitory effects against AChE, creating a strong neuroprotective basis. A few medicinal herbs, including PG, CL, CA, GB, *Glycyrrhiza species*, WS, BM, TC, and CP, have improved outcomes for AD patients. In vitro and in vivo trials using extracts of some of these herbs showed significant cognitive protection, decreased Aβ plaque formation, prevention of neuronal degeneration, and prevention of synaptic loss were also observed.

There are few FDA-approved drugs for treating AD, none of which are natural antioxidant molecules, as few therapeutic agents pass the clinical trial in the in vivo phase. Instead of randomized clinical studies, experimental antioxidant therapies have shown promising outcomes in AD animal models. However, bioactive molecules have a wide variety of chemical components and incredibly low toxicity; as a result, AD prevalence has decreased globally. Additionally, examining the antioxidant properties of natural products as AD therapeutic interventions is essential. Neuroscience is still in its infancy when considering bioactive compounds in the context of AD. However, these substances can control ROS generation for a range of cellular mechanisms brought on by AD progression, which may result in the development of a newer and more potent therapeutic approach and a more practical way to reduce the onset of AD.

**Author Contributions:** K.N. and M.P.: conceptualization and design review and review final version approval. K.N. and K.T.N.: bibliographic research. K.N. and M.P.: writing—original draft preparation. A. and K.Y.: table and figure design. M.P.: supervision. All authors have read and agreed to the published version of the manuscript.

**Funding:** This research received no external funding.

**Institutional Review Board Statement:** Not applicable.

**Informed Consent Statement:** Not applicable.

**Data Availability Statement:** Not applicable.

**Acknowledgments:** The authors want to acknowledge the facilities provided by the Rungta College of Pharmaceutical Sciences and Research, Kohka, Kurud Road, Bhilai, Chhattisgarh, India. This research did not receive any specific grant from funding agencies in the public, commercial, or not-for-profit sectors.

**Conflicts of Interest:** The authors declare no conflict of interest.

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
