# Peer review of "Unlocking the Therapeutic Potential of Medicinal Plants for Alzheimer’s Disease: Preclinical to Clinical Trial Insights"

_futurepharmacol, doi:10.3390/futurepharmacol3040053_

Round 1
Reviewer 1 Report
Comments and Suggestions for Authors
The aim of the study was to give an overview of the effectiveness of medicinal plants in treating Alzheimer’s disease. In the literature, there are several review papers on this topic and this manuscript does not offer a special upgrade to existing work. The authors briefly explained some of the main hypotheses related to AD development, gave an overview of the AD-related effects of the extracts from several promising plants, and listed AD clinical trials involving herbal remedies. In my opinion, the manuscript presented has several limitations.
General description of pathophysiological mechanisms driving the onset and progression of AD is too simple. Mechanisms such as oxidative stress, neuroinflammation, autophagy and neuronal death are not explained at all, which is important as effects of many of medicinal plants are related to oxidative stress and neuroinflammation. Presentation of Aβ and tau pathology-related mechanisms is outdated, recent discoveries (tau truncation, disease spreading,…..) should be at least mentioned. This part of the manuscript needs detailed upgrade.
Part of the manuscript related to medicinal plants is relatively good, although more plants could have been included in the text section.
Part of the manuscript related to clinical trials is not informative. The authors listed clinical trials performed but did not present results of these studies, which is essential if the focus of the manuscript is on the pharmacological potential of medicinal plants.
English should be improved, too many sentences are difficult to follow (some examples are listed below)
Some ref. are misused. In general, more original studies should be cited to support medicinal plants as promising anti-AD therapy.
Word template should be improved.
Table 1 – in general, column Biological activity could be omitted, the table should be focused on AD-relevant findings; the authors put the same effects in different columns (Mode of anti-Alzheimer action and Biological activity), some examples are listed below but the whole table should be corrected according to these suggestions; all citations should refer to original articles; for most of the plant listed there are more anti-alzheimer activity than indicated in the table
1. Cholinergic activity should be added to Mode of anti-Alzheimer action
2. Glycyrrhiza uralensis – should be deleted and included into 4.5
5. AChE inhibitory potential -put into Mode of anti-Alzheimer action
6. anti-A? aggregation, anti-neuroinflammatory properties, and AChE inhibitory activity – put into Mode of anti-Alzheimer action
7. should be deleted and included into 4.4
8. memory impairment and amelioration of cognitive deficits – redundancy
9. should be included in 4.2
13. ref. 144 is related to hepato-renal toxicity
18. Lavender – no evidence for anti-alzheimer mode of action in ref. 151
Table 2: Summary of clinical trial conducted for anti-Alzheimer medicinal plants – RESULTS should be added
Other specific comments are listed bellow:
Page 2 – “More than 145 percent of deaths attributed to AD were observed between 2000 and 2019” – unclear
Page 2 - as about 850,000 people living with dementia – please specify the year
Page 2 – “the most frequent cause of AD in older people is dementia” – the relation between AD and dementia should be explained more clearly, mechanistically, dementia is not cause of AD
Epidemiological data regarding dementia could be omitted, the paper should be focused on AD
Page 2 – “AD is a neuronal cell disorder that generates progressive dysfunction of neurons due to the deposition of protein, genetic predispositions, and increased oxidative stress” – too simplified description of mechanisms underlying the neuronal dysfunction
Page 2 -To date, there is no disease-modifying or neuroprotective medication available at this time. – duplication
Page 2 – “acetylcholinesterase inhibitors (AChEI), a class of drugs created especially for Alzheimer's, began in the late 1990s [9].” – please cite reference relevant for AChEI not tau; inappropriate citation also for ref. 14, 27, 83, 85
Page 2 – “consumers believed that herbal medicines are cost-effective and more effectively treat some ailments than traditional therapies [18,19].” – it is not important what consumers believe
Page 3 – “Plant-derived natural products are a more appealing source of biologically active molecules than chemical synthesis since they are organic and cost-effective [22,23].” – should be explained clearly, plant-derived natural products contain chemical organic compounds
Page 3 – “in increasing effectiveness or combating AD.” – what is meant by increasing effectiveness
Page 3 – “AD seems to be the most prevalent cause of dementia” – on page 2 the authors wrote that dementia is cause of AD
Figure 1 is incomplete, oxidative stress in not mentioned at all, the figure too much resembles to figures characteristic for multiple sclerosis from the literature
Page 4 – “APP cleaves the Aβ and releases it to the exterior and is later degraded, in aging patients, due to the lower metabolic activity of degraded Aβ, it gets accumulated.” – APP does not cleave Aβ, accumulation of Aβ is not explained correctly, what is meant by lower metabolic activity of degraded Aβ?, BACE1 should be included in the explanation
Page 4 – “amyloid fibrils, which upon accumulation, cause SPs to form and the induction of tau..” – the relationship between Aβ and tau is much more complex, reasons for tau accumulation are not explained well
Page 5 – “The fact that many patients have amyloid plaque burden but no cognitive impairment symptoms suggested that the hypothesis may not be totally accurate [13,31,32].” – I agree, but this is in contrast to page 3 where authors wrote that “(Aβ)-peptides play a crucial role in the pathophysiology of AD”
Page 5 – what is meant by “drugs that can target many aspects of the disease, such as …..mechanism-based versus non-mechanism-based strategies”?
Page 5 – “They are essential for maintaining the neuronal neurotransmitters in the brain by reducing the effects of various receptors through different mechanisms [41].” – if they are essential, than it is impossible to keep normal levels of neurotransmitters without natural products which is not correct, please write accurately
Page 6 – “Medicinal plants that have been shown to be effective in treating AD includes Panax ginseng (PG), Curcuma longa (CL), Centella asiatica (CA), Ginkgo biloba (GB), Glycyrrhiza glabra, Withania somnifera (WS), Bacopa monnieri (BM), Tinospora cordifolia (TC), and Convolvulus pluricaulis (CP) have been exclusively discussed in this review, which is also depicted in the Fig.2, indicating the mechanism of these medicinal plants which further providing a new insight for the treatment of AD [13].” – correct the sentence
Figure 2 implies that each plant has only one activity, the Figure should be reorganized indicating that compounds from each plant may display various effects; on large magnification font is not clearly visible
Page 7 – ginseng oils is not class of chemical compounds
Page 8 - makes this constituent? plural
Page 8 – “Additionally, it has been found that ginsenoside lowers β--secretase activity, reduces calcium influx and exhibits a regulatory influence on the PI3K/AKT pathway, thereby lowering the Tau protein's phosphorylation.” – importance of calcium ions and PI3K/Akt pathway for AD should be introduced before this part
Page 8 - anti-oxidative stress?
Page 8 – “lysophosphatidic acid 1/3 receptors (LPA1/3)” – should be better combined with the text few lines above to avoid repetition
Page9 – “that proteins related to mitochondria and synapse” – disturbance of mitochondrial functions and synaptic loss in AD should be introduced previously
Page 9 – “against the above-stated disorders [59,60].” – references cited are related to AD only
Page 9 – “In AD, deregulation of Cyclin-dependent kinase 5 (Cdk5) activity by the production of its hyperactivator p25, leading to the formation of tau and amyloid protein, is the key pathological feature of AD.” – I suggest important instead of key
Page 10 - abnormal alternation behaviour?
Page 10 – “preventing the activation of the NLRP3 inflammasome in addition to stimulation of microglial M2 polarisation.” – NLRP3 inflammasome and microglial M2 polarisation should be explained previously
Page 10 – “to elucidate its SAR studies” - ?
Page 11 – “In addition, CA contains centellosides (pentacyclic triterpenoid saponins) which is considered to be one of the main bioactive compound of the plant [84]” – plural/are
Page 11 –“ with assuring pharmacological activities” - ?
Page 11 – “CNS disorders such as epilepsy, dementia, alzheimer, and anxiety. Among these, the CA extract has been shown to exhibit promising improvement in learning and memory in AD patients [85,86]. Research reports have proven CA to be a potential anti-AD herbal drug. Crude extracts and pure phytoactives of CA have successfully managed AD in pre-clinical and clinical animal models.” – repetition
Page 12 - amyloid-b1e40 (Ab40) and methionine amyloid-b1e40 (MAb40) fibrillation?
Page 12 - A1-42 (50 M)?
Page 12 - p NFkB p65, NF-B p65 – please correct
Page 13 – “neurodegenerative disorders such as dementia” – this should be specified more clearly
Page 13 – “immunopathogenic events of AD, for instance, inhibition of pTau and Aβ inhibition.” – accumulation of pTau and Ab are not immunopathogenic events
Page 13 – “has been reported to inhibited by phytoactive compound present in Glycyrrhiza species” - ?
Page 13 – “Moreover, when compared to the scopolamine-treated group, the superoxide dismutase activity, acetylcholinesterase activity, and catalase activity in the glycyrrhizic acid-treated group significantly reversed cognitive impairment.” – should be written more clearly
Check ref.105
Page 14 – “The experimental data suggested that enhancement in fibril formation is seen when bacoside-A and PrP(106–126) are co-incubated in the presence of vesicle bilayers while in parallel disrupt membrane bilayers of the peptide assemblies.” – unclear
Page 15 – “One study suggested that the metabolites isolated from TC potentially contribute to AD.” – contribute to AD?
Page 15 – “the findings of 8-hydroxytinosporide with standard donepezil” – unclear
Page 15 – “Another finding suggested that supplementing TC to the person suffering from behavioural issues and memory impairment may improve thinking skills and memory [112].” – misleading, please cite the clinical data supporting this claim
Page 16 – “The study uncovers the effects of CP extract (CPE) on a chronic rat model of depression, and the findings revealed that in the (chronic unpredictable mild stress) CUMS-exposed rats, fluoxetine (10 mg/kg, p.o.) or CPE (50, and 100 mg/kg) treatments given back-to-back over one week markedly increased sucrose preference index, decreased immobility time in the FST, and increased the number of squares crossed, and locomotion in the actophotometer” – how are these tests related to AD? Please explain what can be concluded in relation to AD.
Page 16 – “These findings imply that CP may be used to treat neurodegeneration caused by stress [117].” – why is study about stress included in AD manuscript?
Page 17 – what is meant by “active site of (Aβ25-35)”?
Page 17 – “The cytotoxic activity carried out on the human SK N-SH cell line reported IC 50 value to be 28.61 ± 2.91, 14.84 ± 1.45, 18.76 ± 0.76, and 30.14 ± 2.59 μM, respectively.” – these values indicate what?
Page 17 – “Moreover, it appears from 100 ns simulations that the withanolide B and withanoside V complexes were stable inside the hydrophobic core of β-amyloid [127].” – please explain why is this stability important and why is mentioned
Page 5, line 16 – medical uses is not appropriate term
Page 5, line 16 – “However, 26 bioactive molecules have a wide variety of chemical components and incredibly 27 low toxicity; as a result, AD prevalence has increased globally.” – it is not clear how is variety of chemical components related to increased AD prevalence
Comments on the Quality of English Language
Comments on the quality of English Language are included in the main report.
Author Response
"Please see the attachment."

Reviewer 2 Report
Comments and Suggestions for Authors
The paper is well-documented and interesting. it bring good amounts of information in the pharmacognosy of traditional medicinal plants at it might be a welcome addition to the knowledge in the field.
The etiological and pathophysiological introduction is welcome, but it should be enhanced comparatively to the mechanisms discussed in the presumed action mechanisms of the various plants, as it seems disjointed and not very helpful after we start to read about the particular actions of each plant. Also, a graphical aid like a schematic of the pathology of the major neurodegenerative diseases would be welcome. Nice graph no 2, interesting and illustrative.
However, writing is sloppy, the writer has very limited knowledge of scientific English and this sometimes makes the test difficult to comprehend. Extensive editing and rewriting is necessary. As it was a nice and interesting lecture, I will attach a proofed copy of the document, with the changes to be made. Otherwise a good text is lost. Please use also an automatic speller or a newer AI-run text enhancers, that will markedly improve the writing.

The paper is well-documented and interesting. it bring good amounts of information in the pharmacognosy of traditional medicinal plants at it might be a welcome addition to the knowledge in the field.
However, writing is sloppy, the writer has very limited knowledge of scientific English and this sometimes makes the test difficult to comprehend. Extensive editing and rewriting is necessary. As it was a nice and interesting lecture, I will attach a proofed copy of the document, with the changes to be made. Otherwise a good text is lost. Please use also an automatic speller or a newer AI-run text enhancers, that will markedly improve the writing.
Round 2
Reviewer 1 Report
Comments and Suggestions for Authors
The authors greatly improved their manuscript. However, there are still some concerns that are mainly related to English. Besides, the authors should carefully read their manuscript to avoid duplication and repetition. Furthermore, in Table 2 - Summary of clinical trials, results of clinical trials are still missing, they must be added if considering medicinal herbs as potential pharmacological intervention (ongoing or completed is not the result of a study).
The authors wrote: The most prevalent form of dementia worldwide is AD (page 1) and Furthermore, the most common cause of dementia is AD (page 2). This is not the same. The authors should clearly indicate the relationship between dementia and AD.
Page2: More than 145 percent of deaths – unclear, what is meant by 145% of death?
projected to be come? - English
Page2: - duplication
As the world's population ages, the number of people with dementia is currently projected to be come over 40 million, and this number is expected to keep growing, doubling every 20 years.
Moreover, around 58 million individuals worldwide are estimated to be suffering from AD as of 2021, which is anticipated to increase to 88 million by 2050. The survey data also reveals that approximately 6.2 million people in United States are suffering from AD [5], [6].
Page 2: - abbreviations should be explained
Tau is primarily expressed in neurons and is significantly regulated by a variety of PTMs. Tau aggregation and accumulation can be caused by abnormal PTMs, LLPS, and pathogenic tau seeds via many pathways.
Page 3:
A wide range of medicinal plants, such as Salvia triloba, Melissa officinalis, Ca-mellia sinensis, Citrus reticulata, and their bioactive plant compounds and crude extracts, seem to have significant potential to treat and pre-vent AD [22], [23]. – these plants are not discussed later, why are they mentioned?
Page 5:
Alzheimer's disease is characterised neuropathologically by Neuritic plaques in the brain. They are produced as a result of extracellular amyloid-β protein (Aβ) deposition and…. - abbreviation for Aβ is introduced previously
Aβ is derived from the sequential cleavage of amyloid-β precursor protein (APP) – the same for APP
It is not clear if either abeta or tau pathology is the first event
Page 4: the increase of amyloid-β (Aβ) is the key event in AD that triggers tau pathology followed by neuronal death and eventually, the disease
Page 5: According to the tau hypothesis, tau tangle pathology develops prior to the formation of Aβ plaque
Page 13: what is SAR activity?
Comments on the Quality of English LanguageManuscript is improved, but it is still not of good quality. English is particularly problematic. The manuscript cannot be accepted without professional English editing.
Reviewer 2 Report
Comments and Suggestions for Authors
Much improved. Page twenty "tunnel assay showed reduced..."
Otherwise duly rewritten.
Comments on the Quality of English LanguageVery much improved
Round 3
Reviewer 1 Report
Comments and Suggestions for Authors
For better clarity please change "More than 145 percent of
deaths attributed to AD......." to An increase of more than 145% for deaths attributed to AD were observed between 2000 and 2019"
Table 2: results - good job!, please improve visual design of the Table
Comments on the Quality of English LanguageModerate editing of English language required
